# Distributed Learning without Distress: Privacy-Preserving Empirical Risk Minimization

**Bargav Jayaraman**
Department of Computer Science
University of Virginia
Charlottesville, VA 22903
bj4nq@virginia.edu

**Lingxiao Wang**
Department of Computer Science
University of California, Los Angeles
Los Angeles, CA 90095
lingxw@cs.ucla.edu

**David Evans**
Department of Computer Science
University of Virginia
Charlottesville, VA 22903
evans@virginia.edu

**Quanquan Gu**
Department of Computer Science
University of California, Los Angeles
Los Angeles, CA 90095
qgu@cs.ucla.edu

## Abstract

Distributed learning allows a group of independent data owners to collaboratively learn a model over their data sets without exposing their private data. We present a distributed learning approach that combines differential privacy with secure multi-party computation. We explore two popular methods of differential privacy, output perturbation and gradient perturbation, and advance the state-of-the-art for both methods in the distributed learning setting. In our output perturbation method, the parties combine local models within a secure computation and then add the required differential privacy noise before revealing the model. In our gradient perturbation method, the data owners collaboratively train a global model via an iterative learning algorithm. At each iteration, the parties aggregate their local gradients within a secure computation, adding sufficient noise to ensure privacy before the gradient updates are revealed. For both methods, we show that the noise can be reduced in the multi-party setting by adding the noise inside the secure computation after aggregation, asymptotically improving upon the best previous results. Experiments on real world data sets demonstrate that our methods provide substantial utility gains for typical privacy requirements.

## 1 Introduction

In many applications, such as medical research and financial fraud detection, it is valuable to build machine learning models by training on sensitive data. This raises privacy concerns since adversaries may be able to infer information about the training data from the learned model. Model parameters can reveal sensitive information about individual records including specific features of the records [20] to the presence of particular records in the data set [47]. In the case of neural networks, the model parameters can also inadvertently store sensitive parts of the training data [8]. Differential privacy [19, 16] aims to thwart such analysis. It provides statistical privacy for individual records by adding random noise to the model parameters. Many works have shown that differential privacy can be used to enable privacy-preserving machine learning in the centralized setting where a single organization owns all the data [10, 11, 29, 30, 51, 58].

The problem becomes more acute when the data is owned by different organizations that wish to collaboratively learn from their private data. For instance, multiple hospitals may want to collabora-

tively train a classifier over their patient medical records without disclosing their own records to other hospitals. The goal of distributed machine learning (also referred to as *federated learning* [36]) is to enable a group of independent data owners to develop a model from their combined data without exposing that data to others.

Multi-party computation (MPC) protocols allow participants to jointly compute a functionality over their private inputs by employing cryptographic techniques like homomorphic encryption, secret sharing, and oblivious transfer. Lindell and Pinkas [31] proposed one of the earliest approaches to use MPC for private data mining, which was followed by several works considering different adversarial models or applications [55, 50, 33, 42]. A recent focus has been to achieve practical and efficient distributed machine learning using MPC protocols [12, 52, 34], and in certain settings such methods have been shown to scale to learning tasks with hundreds of millions of records [39, 21]. However, unlike approaches using differential privacy on the model, these approaches only protect the training data during the learning process; they provide no protection against inference attacks on the resulting model.

Pathak et al. [41] proposed the first differentially-private machine learning in distributed setting. Their method securely aggregates local models and uses output perturbation to achieve differential privacy. However, the noise scales inversely proportional to the smallest data set size of the $m$ parties. This can be improved by a factor of $\sqrt{m}$ by first training differentially-private local models using the method of Chaudhuri et al. [11], and then performing naïve aggregation of the local models. In this work, we propose an output perturbation method that improves over Pathak et al.'s method by a factor of $m$ by adding the noise inside an MPC with the scale of noise required roughly inversely proportional to the size of the entire data set. Recent works on distributed noise generation [17, 3, 24, 45] try to achieve a similar bound by requiring parties to add partial noise locally, and combining these noises to ensure differential privacy. However, these methods require additional noise to tolerate corruptions and collusion. More concretely, with a minimum of $k$ honest parties out of $m$, their noise bound is worse than ours by a factor of $\sqrt{m/k}$. On the other hand, in our approach the noise is generated inside the MPC such that any honest participant can be assured that sufficient noise is added to protect their own privacy even if all other participants are dishonest and colluding.

While these model aggregation approaches are computationally efficient, they tend to produce less accurate global models compared to the centralized setting, especially when the number of data owners is large (in the extreme, when each party has only one training instance). For such scenarios, distributed iterative learning with gradient perturbation is a better option. Shokri and Shmatikov [46] provide such a solution for deep learning, where the local gradients are perturbed and then revealed for updating the global model. Their privacy budget is *per parameter*, however, and not for the entire training so huge total privacy budgets are required. Abadi et al. [1] proposed a tighter bound on the privacy budget using moments accountant which is applicable to centralized setting. Wang et al. [51] used the moments accountant to propose iterative learning with gradient perturbation for the centralized setting. We propose a method for distributed setting using *zero-concentrated differential privacy* [6] which achieves similar tight bound on privacy budget. Moreover, we add noise inside MPC after gradient aggregation, thus reducing the noise by a factor of $\sqrt{m}$ compared to the naïve aggregation of noisy gradients. While Chase et al. [9] also achieve similar bound on noise in distributed learning setting, their method considers only the convex case. We achieve a different (and tighter) utility bound for the strong convexity case. Further, Chase et al. use differential privacy which has different composition properties than the zero-concentrated differential privacy that we consider. We also note that the method proposed by Rajkumar and Agarwal [43] has similar objectives, but their protocol requires a trusted third party to execute the SGD algorithm, whereas our method does not depend on any trusted party. In addition, although their method has the same scale of noise as ours, in their method each party samples local noise which is aggregated by the trusted third party. This is not secure in the presence of colluding parties as noted by Bindschaedler et al. [3] and Shi et al. [45]. In our method, parties collaboratively generate noise within the MPC. Finally, their method requires noise from two sources: the Gaussian noise $\eta$ and the Laplace noise $\rho$. Generation of $\rho$ consumes $\epsilon$ privacy budget per iteration, as opposed to using $\epsilon$ budget for the entire learning process, and hence violates the privacy constraints.

In this paper, we introduce differentially-private distributed machine learning protocols using both output perturbation and gradient perturbation where the noise is added within a secure multi-party computation. Our output perturbation method securely aggregates the local models and achieves $\epsilon$-differential privacy by adding Laplace noise to the aggregated model parameters. In our gradient

perturbation method, the parties collaboratively run an iterative, gradient-based learning algorithm where they securely aggregate the local gradients at each iteration. This provides $(\epsilon, \delta)$-differential privacy by adding Gaussian noise to the aggregated gradients. In both the methods the sampled noise is (roughly) inversely proportional to the size of the entire data set. While the first method is computationally efficient, requiring only single invocation of MPC, its accuracy decreases compared to centralized method when the number of parties is large relative to the total amount of data — this is inherent to any model aggregation based method. The iterative gradient perturbation method, on the other hand, does not suffer from accuracy degradation but requires one MPC protocol execution per iteration. Both methods achieve accuracy close to their non-private counterparts where no noise is added and no data privacy provided.

## 1.1 Contributions

This work makes the following contributions, which address challenges in distributed learning.

**Output Perturbation and Gradient Perturbation Methods.** We propose two approaches to privately train accurate machine learning models in the distributed setting. While the output perturbation method (Section 3.1) is computationally more efficient, the gradient perturbation method (Section 3.2) maintains high accuracy regardless of how the data is partitioned.

**Reduced Noise Bounds.** We give noise bounds for each method that are smaller than the best previous approaches for output perturbation [41] and gradient perturbation [46], while ensuring differential privacy in the distributed setting (Theorem 3.1 for output perturbation and Theorem 3.4 for gradient perturbation). For gradient perturbation, we use *zero-concentrated differential privacy* to achieve the lowest known bound on the privacy budget. Moreover, we generate the noise within the MPC protocol. This allows us to add only a single copy of noise, compared to previous works that combine noise from each participant [17, 4, 3, 24, 45]. We provide a theoretical analysis of our methods' error bounds which match the state-of-art error bounds in centralized settings.

**Experimental Evaluation on Real Data Sets.** We implement regularized logistic regression and regularized linear regression models for classification and regression tasks respectively. We report results from experiments performed on the KDDCup99 and Adult data sets for classification and the KDDCup98 data set for regression. We compare our methods with previous work on distributed learning, varying the number of parties and local data set sizes. Our methods produce models that are closest to the non-private models in terms of model accuracy and generalization error since we add less noise than previous distributed learning methods.

## 2 Background on Differential Privacy and Multi-Party Computation

This section introduces differential privacy (including the zero-concentrated differential privacy notion we use), and secure multi-party computation.

**Notation:** For any $d$-dimensional vector $\mathbf{x} = [x_1, ..., x_d]^\top$, we use $\|\mathbf{x}\| = (\sum_{i=1}^{d} |x_i|^2)^{1/2}$ to denote its $\ell_2$-norm. Given two sequences $\{a_n\}$ and $\{b_n\}$, we write $a_n = O(b_n)$ if there exists a constant $0 < C < \infty$ such that $a_n \leq C b_n$, and we use $\widetilde{O}(\cdot)$ to hide the logarithmic factors.

### 2.1 Differential Privacy

Differential privacy was introduced by Dwork [18] and is defined as follows:

**Definition 2.1** (($\epsilon, \delta$)-Differential Privacy)**.** Given two adjacent data sets $D, D' \in \mathcal{D}^n$ differing by a single element, a randomized mechanism $\mathcal{M} : \mathcal{D}^n \rightarrow \mathbb{R}^d$ provides $(\epsilon, \delta)$-differential privacy if it produces response in the set $S$ with probability $\mathbb{P}[\mathcal{M}(D) \in S] \leq e^\epsilon \mathbb{P}[\mathcal{M}(D') \in S] + \delta$.

The above definition reduces to $\epsilon$-Differential Privacy ($\epsilon$-DP) when $\delta = 0$. We can achieve $\epsilon$-DP and $(\epsilon, \delta)$-DP by adding noise sampled from Laplace and Gaussian distributions respectively, where the noise is proportional to the sensitivity of $\mathcal{M}$, given as $\Delta\mathcal{M} = \|\mathcal{M}(D) - \mathcal{M}(D')\|$.

Throughout this paper we assume the $\ell_2$-sensitivity which considers the upper bound on the $\ell_2$-norm of $\mathcal{M}(D) - \mathcal{M}(D')$.

**Zero-Concentrated Differential Privacy.** While, the notion of differential privacy performs well for methods like output perturbation, it is not suitable for gradient perturbation methods which require repeated sampling of noise in the iterative training procedure. Zero-concentrated differential privacy [6] (zCDP) has a tight composition bound and hence is a better option for gradient perturbation.

We first define the privacy loss random variable which is used in the definition of zCDP.

**Definition 2.2.** For two adjacent data sets $D, D' \in \mathcal{D}^n$ differing by one sample, a randomized mechanism $\mathcal{M} : \mathcal{D}^n \to \mathbb{R}^d$, and an outcome $o \in \mathbb{R}^d$, the privacy loss random variable $Z$ is defined as

$$Z = \log \frac{\mathbb{P}[\mathcal{M}(D) = o]}{\mathbb{P}[\mathcal{M}(D') = o]}. \tag{1}$$

**Definition 2.3.** A randomized mechanism $\mathcal{M} : \mathcal{D}^n \to \mathbb{R}^d$ satisfies $\rho$-zCDP if for any two adjacent data sets $D, D' \in \mathcal{D}^n$ differing by one sample, it holds that for all $\alpha \in (1, \infty)$,

$$\mathbb{E}\big[e^{(\alpha-1)Z}\big] \leq e^{(\alpha-1)\alpha\rho}. \tag{2}$$

Note that (2) implies that $\mathbb{P}[Z > \lambda + \rho] \leq e^{-\lambda^2/(4\rho)}$ for all $\lambda > 0$, which suggests that the privacy loss $Z$ is tightly concentrated around zero mean, and hence it is unlikely to distinguish $D$ from $D'$ given their outputs.

Bun and Steinke [6] give the following lemmas to achieve zCDP with the Gaussian mechanism. Lemma 2.1 bounds the amount of Gaussian noise to guarantee $\rho$-zCDP. Lemma 2.2 gives the composition of multiple zCDP mechanisms. Finally, Lemma 2.3 specifies the mapping from $\rho$-zCDP to $(\epsilon, \delta)$-DP.

**Lemma 2.1.** *Given a function $q : \mathcal{D}^n \to \mathbb{R}^d$, the Gaussian Mechanism $\mathcal{M} = q(D) + \mathbf{u}$, where $\mathbf{u} \sim N(0, \sigma^2 \mathbf{I}_d)$, satisfies $\Delta_2(q)^2/(2\sigma^2)$-zCDP.*

**Lemma 2.2.** *For two randomized mechanisms $\mathcal{M}_1 : \mathcal{D}^n \to \mathbb{R}^d$, $\mathcal{M}_2 : \mathcal{D}^n \times \mathbb{R}^d \to \mathbb{R}^d$. If $\mathcal{M}_1$ satisfies $\rho_1$-zCDP and $\mathcal{M}_2$ satisfies $\rho_2$-zCDP, then $\mathcal{M}_2(D, \mathcal{M}_1(D))$ satisfies $(\rho_1 + \rho_2)$-zCDP.*

**Lemma 2.3.** *If a randomized mechanism $\mathcal{M} : \mathcal{D}^n \to \mathbb{R}^d$ satisfies $\rho$-zCDP, then it satisfies $(\rho + 2\sqrt{\rho \log(1/\delta)}, \delta)$-differential privacy for any $\delta > 0$.*

## 2.2 Secure Multi-Party Computation

Our threat model considers semi-honest participants who wish to compute a joint functionality without revealing their individual inputs to other participants. In this threat model, while the parties do not tamper with the joint functionality or provide garbage inputs, they are allowed to passively infer about inputs of other parties based on the protocol execution. We use generic multi-party computation protocols to securely aggregate local models and gradients. A multi-party computation (MPC) protocol enables two or more parties to jointly compute a function of their private inputs, without disclosing any information about those inputs other than their size and whatever can be inferred from the revealed output [56]. The notion of MPC goes back to a series of talks given by Andrew Yao in the 1980s. The protocol he introduced, now known as Yao's garbled circuits protocol, can compute any function securely. Numerous other secure multi-party computation protocols have been devised since then (e.g., [22, 32, 14, 37]), and many tools have been developed for efficiently implementing MPC computations (e.g., [35, 13, 7, 27, 26, 44, 53, 57]). It is now practical to execute two-party protocols with millions of inputs [21, 23], and global-scale, many-party protocols with malicious level security for small inputs [54].

Secure aggregation of local classification models using MPC was shown to be practical by Tian et al. [49]. This work used a two-party computation, with a semi-honest threat model and non-colluding servers. A similar approach has been used to scale multi-party regressions [38, 21]. We can use these methods to achieve secure aggregation. For scenarios where the risks of collusion are too high, many-party MPC protocols can be used that provide security to a single honest participant even if all other participants are malicious. In this work, we do not focus on improving or evaluating the MPC execution, since the methods we propose can be implemented using well known MPC techniques. Appendix C provides information on the MPC implementation we use and its cost.

# 3 Multi-Party Machine Learning

In this section we describe our output perturbation and gradient perturbation methods in detail along with theoretic analysis of differential privacy and generalization error bound.

We consider the following empirical risk minimization (ERM) objective:

$$J_D(\theta) = \frac{1}{n} \sum_{i=1}^{n} \ell(\theta, x_i, y_i) + \lambda N(\theta),$$

where $\ell(\theta)$ is a convex loss function that is $G$-Lipschitz and $L$-smooth over $\theta \in \mathbb{R}^d$. $N(\cdot)$ is regularization term. We consider $J(\cdot)$ to be $\lambda$-strongly convex. Each data instance $(x_i, y_i) \in D$ lies in a unit ball. For a party $j$, with data set $D_j$ of size $n_j$, we denote its data instance as $(x_i^{(j)}, y_i^{(j)})$.

## 3.1 Model Aggregation with Output Perturbation

We extend the differential privacy bound of Chaudhuri et al. [10] to the multi-party setting, ensuring sufficient noise to preserve the privacy of each participant's data throughout the multi-party computation, including the final output.

Given $m$ parties, each having a data set $D_j$ of size $n_j$ and the corresponding local model estimator $\widehat{\theta}^{(j)}$ obtained by minimizing the local objective function: $J_{D_j}(\theta) = \frac{1}{n_j} \sum_{i=1}^{n_j} \ell(\theta, x_i^{(j)}, y_i^{(j)}) + \lambda N(\theta)$. The perturbed aggregate model estimator is given as $\widehat{\theta}^{\text{priv}} = \frac{1}{m} \sum_{j=1}^{m} \widehat{\theta}^{(j)} + \eta$, where $\eta$ is the Laplace noise added to the aggregate model estimator to preserve differential privacy. Secure model aggregation can be performed using the framework of Tian et al. [49] as mentioned in Section 2.2.

The next theory provides a bound on the noise magnitude needed to achieve differential privacy:

**Theorem 3.1.** *Given a perturbed aggregate model estimator $\widehat{\theta}^{priv} = \frac{1}{m} \sum_{j=1}^{m} \widehat{\theta}^{(j)} + \eta$ where $\widehat{\theta}^{(j)} = \arg\min_\theta \frac{1}{n_j} \sum_{i=1}^{n_j} \ell(\theta, x_i^{(j)}, y_i^{(j)}) + \lambda N(\theta)$ and the data lie in a unit ball and $\ell(\cdot)$ is $G$-Lipschitz , then $\widehat{\theta}^{priv}$ is $\epsilon$-differentially private if*

$$\eta = Lap\left(\frac{2G}{m n_{(1)} \lambda \epsilon}\right),$$

*where $n_{(1)}$ is the size of the smallest data set among the $m$ parties, $\lambda$ is the regularization parameter and $\epsilon$ is the differential privacy budget.*

*Proof.* Let there be $m$ parties such that one record of party $j$ changes in the neighbouring data sets, then

$$\frac{\Pr(\widehat{\theta}|D)}{\Pr(\widehat{\theta}|D')} = \frac{\Pr\left(\frac{1}{m} \sum_{i \neq j} \widehat{\theta}^{(i)} + \frac{1}{m} \widehat{\theta}^{(j)} + \eta | D\right)}{\Pr\left(\frac{1}{m} \sum_{i \neq j} \widehat{\theta}^{(i)} + \frac{1}{m} \widehat{\theta}'^{(j)} + \eta | D'\right)} = \frac{\exp\left[\frac{m.n_{(1)}\epsilon\lambda}{2G} \frac{\|\widehat{\theta}^{(j)}\|}{m}\right]}{\exp\left[\frac{m.n_{(1)}\epsilon\lambda}{2G} \frac{\|\widehat{\theta}'^{(j)}\|}{m}\right]}$$

$$\leq \exp\left[\frac{n_{(1)}\epsilon\lambda}{2G} \|\widehat{\theta}^{(j)} - \widehat{\theta}'^{(j)}\|\right] \leq \exp\left[\frac{n_{(1)}\epsilon\lambda}{2G} \frac{2G}{n_j \lambda}\right] \leq \exp(\epsilon),$$

where the second inequality follows from Corollary 8 of Chaudhuri et al. [11]. $\square$

We now provide a bound on the excess empirical risk and true risk similar to Pathak et al. [41]. Our bounds are tighter than Parthak et al.'s as we require less differential privacy noise.

**Theorem 3.2.** *Given a perturbed aggregate model estimator $\widehat{\theta}^{priv} = \frac{1}{m} \sum_{j=1}^{m} \widehat{\theta}^{(j)} + \eta$ where $\widehat{\theta}^{(j)} = \arg\min_\theta \frac{1}{n_j} \sum_{i=1}^{n_j} \ell(\theta, x_i^{(j)}, y_i^{(j)}) + \lambda N(\theta)$ and an optimal model estimator $\theta^*$ trained on the centralized data such that the data lie in a unit ball and $\ell(\cdot)$ is $G$-Lipschitz and $L$-smooth, then the bound on excess empirical risk is given as:*

$$J(\widehat{\theta}^{priv}) \leq J(\theta^*) + C_1 \frac{G^2(\lambda + L)}{n_{(1)}^2 \lambda^2} \left(m^2 + \frac{d^2 \log^2(d/\delta)}{m^2 \epsilon^2} + \frac{d \log(d/\delta)}{\epsilon}\right),$$

*where $C_1$ is an absolute constant.*

The proof of Theorem 3.2 follows from Pathak et al. [41]. The main difference is that we use the sensitivity bound as $2G/(mn_{(1)}\lambda)$ instead of $2G/(n_{(1)}\lambda)$ and thereby achieve a tighter bound. The full proof is given in Appendix A.1.

**Theorem 3.3.** *Given a perturbed aggregate model estimator $\widehat{\theta}^{priv} = \frac{1}{m}\sum_{j=1}^{m}\widehat{\theta}^{(j)} + \eta$ where $\widehat{\theta}^{(j)} = \arg\min_\theta \frac{1}{n_j}\sum_{i=1}^{n_j}\ell(\theta, x_i^{(j)}, y_i^{(j)}) + \lambda N(\theta)$ and an optimal model estimator $\theta^*$ trained on the centralized data such that the data lie in a unit ball and $\ell(\cdot)$ is G-Lipschitz and L-smooth, then the following bound on true excess risk holds with probability at least $1 - \gamma$:*

$$\mathbb{E}[\widetilde{J}(\widehat{\theta}^{priv})] - \min_\theta \widetilde{J}(\theta) \leq C_1 \frac{G^2(\lambda + L)}{n_{(1)}^2\lambda^2}\left(m^2 + \frac{d^2\log^2(d/\delta)}{m^2\epsilon^2} + \frac{d\log(d/\delta)}{\epsilon}\right) + C_2\frac{G^2\log(1/\gamma)}{\lambda n},$$

*where $n$ is the size of the centralized data set. $\widetilde{J}(\theta) = \mathbb{E}_{x,y}[\ell(\theta, x, y)] + \lambda N(\theta)$, $C_1, C_2$ are absolute constants, and the expectation is taking with respect to the noise $\eta$.*

See Appendix A.2 for the proof of Theorem 3.3. The true excess risk bound in Theorem 3.3 implies that the private output of our algorithm converges to the population optimum at the order of $1/n$.

## 3.2 Iterative Learning with Gradient Perturbation

We consider this centralized ERM objective for $m$ parties, each with a data set $D_j$ of size $n_j$:

$$J_D(\theta) = \min_\theta \frac{1}{m}\sum_{j=1}^{m}\frac{1}{n_j}\sum_{i=1}^{n_j}\ell(\theta, x_i^{(j)}, y_i^{(j)}) + \lambda N(\theta).$$

The parties can collaboratively learn a differentially private model via iterative learning by adding noise to the aggregated gradients within the MPC in each iteration with the following noise bound.

**Theorem 3.4.** *Given a centralized model estimator $\theta_T$ obtained by minimizing $J_D(\theta)$ after $T$ iterations of gradient descent algorithm executed jointly by $m$ parties each having dataset $D^{(j)}$ of size $n_j$ where each data instance $(x_i^{(j)}, y_i^{(j)}) \in D^{(j)}$ lie in a unit ball and $\ell(\theta)$ is G-Lipschitz and L-smooth over $\theta \in \mathcal{C}$. If the learning rate is $1/L$ and the gradients are perturbed with noise $z \in \mathcal{N}(0, \sigma^2 I_d)$, then $\theta_T$ is $(\epsilon, \delta)$-differentially private if*

$$\sigma^2 = \frac{8G^2 T\log(1/\delta)}{m^2 n_{(1)}^2\epsilon^2}, \tag{3}$$

*where $n_{(1)}$ is the size of the smallest data set among the $m$ parties.*

*Proof.* Given a gradient at step $t$,

$$M_t = \nabla J(\theta, D) + \mathcal{N}(0, \sigma^2 I_p) = \frac{1}{m}\sum_{j=1}^{m}\frac{1}{n_j}\sum_{i=1}^{n_j}\nabla\ell(\theta, x_i^{(j)}, y_i^{(j)}) + \mathcal{N}(0, \sigma^2 I_p).$$

We assume that only one data instance of one party changes in neighbouring datasets $D$ and $D'$. Hence the sensitivity bound, $\|\nabla J(\theta, D) - \nabla J(\theta, D')\| \leq \frac{2G}{mn_{(1)}}$.

Thus, using Lemma 2.1, $M_t$ is $\rho$-zCDP with $\rho = \frac{2G^2}{m^2 n_{(1)}^2\sigma^2}$. By composition from Lemma 2.2, we observe that $\theta_T$ is $T\rho$-zCDP. Applying Lemma 2.3, we obtain $T\rho + 2\sqrt{T\rho\log(1/\delta)} = \epsilon$. Solving the roots of this equation, we obtain

$$\rho \approx \frac{\epsilon^2}{4T\log(1/\delta)} \implies \sigma^2 = \frac{8G^2 T\log(1/\delta)}{m^2 n_{(1)}^2\epsilon^2}.$$

Thus, $\theta_T$ is $(\epsilon, \delta)$-differentially private for the above value of $\sigma^2$. $\qquad\qquad\square$

Additionally, we observe that differential privacy is also guaranteed for each intermediate model estimator:

**Corollary 3.5.** *Intermediate model estimator $\theta_t$ at each iteration $t \in [1, T]$ is $(\sqrt{t/T}\epsilon, \delta)$-differentially private.*

Hence, an adversary cannot obtain additional information from the intermediate computations. See Appendix A.3 for the proof of Corollary 3.5.

Next, we provide theoretical bounds on the excess empirical risk and the true excess risk of our proposed method.

**Theorem 3.6.** *Given a centralized model estimator $\theta_T$ obtained by minimizing $J_D(\theta)$ after $T$ iterations of gradient descent algorithm executed jointly by $m$ parties each having dataset $D^{(j)}$ of size $n_j$ where each data instance $(x_i^{(j)}, y_i^{(j)}) \in D^{(j)}$ lie in a unit ball and $\ell(\theta)$ is G-Lipschitz and L-smooth over $\theta \in \mathcal{C}$. If the learning rate is $1/L$ and the gradients are perturbed with noise $z \in \mathcal{N}(0, \sigma^2 I_d)$ with $\sigma^2$ defined in (3), and if we choose the iteration number as*

$$T = \widetilde{O}\left( \log \left( \frac{m^2 n_{(1)}^2 \epsilon^2}{dG^2 \log(1/\delta)} \right) \right),$$

*then we have a bound on excess empirical risk:*

$$\mathbb{E}[J(\theta_T)] - J(\theta^*) \leq C_1 \frac{G^2 L d \log^2(mn_{(1)}) \log(1/\delta)}{m^2 n_{(1)}^2 \lambda^2 \epsilon^2},$$

*where the expectation is taking with respect to the noise $\eta$, $n_{(1)}$ is the size of the smallest data set among the $m$ parties, $C_1$ is an absolute constant.*

Appendix A.4 provides the proof.

Based on the excess empirical risk, we next derive the true excess risk.

**Theorem 3.7.** *Given a centralized model estimator $\theta_T$ obtained by minimizing $J_D(\theta)$ after $T$ iterations of a gradient descent algorithm executed jointly by $m$ parties each having dataset $D^{(j)}$ of size $n_j$ where each data instance $(x_i^{(j)}, y_i^{(j)}) \in D^{(j)}$ lie in a unit ball and $\ell(\theta)$ is G-Lipschitz and L-smooth over $\theta \in \mathcal{C}$. If we choose the learning rate, noise level, and iteration number as suggested in Theorem 3.6, with probability at least $1 - \gamma$, we have the following bound on true excess risk:*

$$\mathbb{E}[\widetilde{J}(\theta_T)] - min_\theta \widetilde{J}(\theta) \leq C_1 \frac{G^2 L d \log^2(mn_{(1)}) \log(1/\delta)}{m^2 n_{(1)}^2 \lambda^2 \epsilon^2} + C_2 \frac{G^2 \log(1/\gamma)}{\lambda n},$$

*where $n$ is the size of the centralized data set, $n_{(1)}$ is the size of the smallest data set among the $m$ parties and $C_1, C_2$ are absolute constants.*

Theorem 3.7 (proof in Appendix A.5) suggests that the output of our iterative gradient perturbation method converges to the population optimum at the order of $1/n$. Note that our true excess risk bound is comparable to that of Wang et al. [51] in centralized setting.

## 4   Experiments

We report on experiments for both classification and regression tasks. For classification, we use a regularized logistic regression model over the KDDCup99 [25] data set (additional experiments on the Adult [2] data set yield similar results, described in Appendix B.3). The KDDCup99 data set contains around 5,000,000 network instances. The task is to predict whether a network connection is a denial-of-service attack or not. We randomly sample 70,000 records and divide it into training set of 50,000 records and test set of 20,000 records. We pre-processed the data according to the procedure of Chaudhuri et al. [11], resulting in records with 122 features. For regression, we train a ridge regression model over the KDDCup98 [40] data set, consisting of demographic and other related information of approximately 200,000 American veterans. The task is to predict the donation amount of an individual in dollars. We randomly sample 70,000 records and divide it into a training set of 50,000 records and test set of 20,000 records. We perform the same pre-processing as in the case of previous data sets and additionally perform feature selection using PCA to retain around 100 features. After pre-processing, each record consists of 95 features.

Table 1: Comparison of noise magnitudes for various multi-party differential privacy methods.

| Pathak | Local Out P | Local Obj P | Local Grad P | MPC Out P | MPC Grad P |
|---|---|---|---|---|---|
| Analytical Bound ($\mathcal{L}$ - Laplace, $\mathcal{N}$ - Gaussian) | | | | | |
| $\mathcal{L}(\frac{2G}{n_{(1)}\lambda\epsilon})$ | $\mathcal{L}(\frac{2G}{\sqrt{m}n_{(1)}\lambda\epsilon})$ | $\mathcal{L}(\frac{2G}{n_{(1)}\epsilon})$ | $\mathcal{N}(\frac{\sqrt{2T}G}{\sqrt{m}n_{(1)}\epsilon})$ | $\mathcal{L}(\frac{2G}{mn_{(1)}\lambda\epsilon})$ | $\mathcal{N}(\frac{\sqrt{2T}G}{mn_{(1)}\epsilon})$ |
| Noise Generation Input ($m = 100$, $n_{(1)} = 500$, $\lambda = 0.01$, $\epsilon = 0.5$, $G = 1$ and $T = 100$) | | | | | |
| $800 \times 10^{-3}$ | $80.0 \times 10^{-3}$ | $8.00 \times 10^{-3}$ | $5.66 \times 10^{-3}$ | $8.00 \times 10^{-3}$ | $0.57 \times 10^{-3}$ |
| Generated Noise (standard deviation over 1000 samples) | | | | | |
| $1150 \times 10^{-3}$ | $112 \times 10^{-3}$ | $11.6 \times 10^{-3}$ | $5.63 \times 10^{-3}$ | $12.2 \times 10^{-3}$ | $0.572 \times 10^{-3}$ |

For all the experiments, we set Lipschitz constant $G = 1$, learning rate $\eta = 1$, regularization coefficient $\lambda = 0.001$, privacy budget $\epsilon = 0.5$, failure probability $\delta = 0.001$ and total number of iterations $T = 1,500$ for gradient descent. We compare our methods with the baselines in terms of optimality gap and relative accuracy loss. Optimality gap is the measure of empirical risk bound $J(\theta) - J(\theta^*)$ over the training data, where $\theta^*$ is the optimal non-private model in the centralized setting. Relative accuracy loss is the difference in the accuracy (mean square error in case of regression) of $\theta$ and $\theta^*$ over the test data. We measure the optimality gap and relative accuracy loss of all the models up to 1,500 iterations of gradient descent training and report the results for different partitioning of training data sets. We vary the number of parties $m$ from 100 (where each party has 500 data instances) to 1,000 parties (with each party having 50 data instances) and up to 50,000 parties (each having only one data instance).

**Baselines for comparison.** For the model aggregation method, we compare to the method of Pathak et al. [41] (denoted as Pathak), and the other differential privacy baselines are obtained by applying the output perturbation (denoted as Local Out P) and objective perturbation (denoted as Local Obj P) method of Chaudhuri et al. [11] on each local model estimator $\widehat{\theta}^{(j)}$ to obtain a differentially private local model estimator and then the model aggregation is performed to obtain the differentially private aggregate model $\widehat{\theta}^{\text{priv}}$. For the iterative learning method, we consider the baseline of aggregation of locally perturbed gradients similar to that of Shokri and Shmatikov [46] (denoted as Local Grad P), except that we improve the noise bound by using zCDP. We also include the method of Rajkumar and Agarwal [43] (denoted as Rajkumar and Agarwal) in our comparison, though note that their method does not provide the same level of privacy as our method. Our output perturbation based model aggregation method and gradient perturbation based iterative learning method are denoted as MPC Out P and MPC Grad P respectively. All the above methods consume a total privacy budget of $\epsilon = 0.5$, except Rajkumar and Agarwal which consumes $\epsilon = 0.5$ budget each iteration. Table 1 summarizes the amount of noise each method needs to preserve differential privacy. As the table shows, our methods add the least amount of noise. Though Local Obj P adds noise in the same range as our methods, it uses the noise in a fundamentally different way. While the other methods add the sampled noise (either via output perturbation or via gradient perturbation) to the optimal non-private model that minimizes the required objective function $J(\theta)$, Local Obj P adds the sampled noise directly to the objective function $J(\theta)$ and hence optimizes an altogether different objective function $J'(\theta) = J(\theta) + \text{Lap}(\frac{2G}{n_1\epsilon})$, which explains why its optimality gap increases with decreasing value of local data set size $n_{(1)}$.

**Results.** Figures 1 and 2 show the results for $m = 1,000$; Appendix B includes plots for other numbers of parties. For both the classification and regression tasks, our proposed methods perform better than the baselines both in terms of optimality gap and relative accuracy loss. For the classification task (Figure 1), MPC Grad P achieves optimality gap in the order of $10^{-3}$ in 500 iterations and relative accuracy loss in the order of $10^{-4}$ within 200 iterations, and MPC Out P also achieves values in the same range. Rajkumar and Agarwal adds noise of the same order as our methods and hence achieves performance close to ours, but as noted earlier, their method consumes $\epsilon$ budget per iteration. Our methods perform order of magnitudes better than the other baselines.

For the regression task (Figure 2), MPC Grad P gradually converges to an optimality gap in the order of $10^{-3}$ and relative accuracy loss in the order of $10^{-2}$. MPC Out P incurs loss due to data partitioning (which is unavoidable even for non-private aggregation methods) but still outperforms the baselines of model aggregation by orders of magnitude.

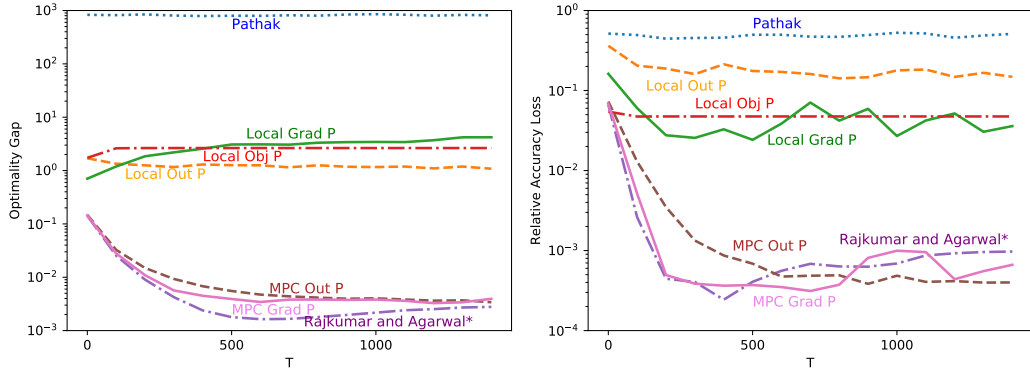

Figure 1: Optimality Gap and Relative Accuracy Loss Comparison on KDDCup99 ($m = 1,000$). (All models have privacy budget $\epsilon = 0.5$, except Rajkumar and Agarwal which consumers $\epsilon = 0.5$ privacy budget each iteration.)

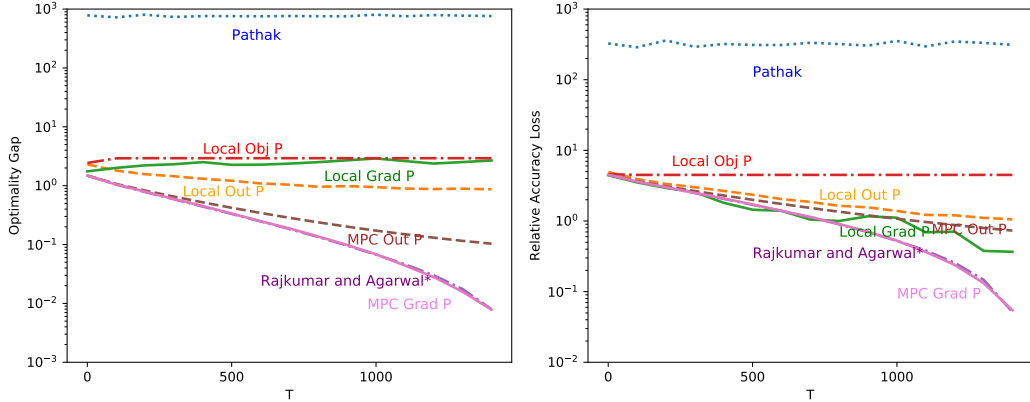

Figure 2: Optimality Gap and Relative Accuracy Loss Comparison on KDDCup98 ($m = 1,000$). (As in Figure 1, all models have privacy budget $\epsilon = 0.5$, except Rajkumar and Agarwal.)

## 5 Conclusions

Our work shows how the noise required for a distributed-learning setting can be reduced by generating and adding noise within a secure computation. Our output perturbation model aggregation method achieves $\epsilon$-differential privacy, and our iterative gradient perturbation method provides $(\epsilon, \delta)$-differential privacy. Both methods improve on the best previous utility bounds for privacy-preserving distributed learning. While our model aggregation method requires only a single secure aggregation (and hence is efficient), our iterative learning method maintains accuracy regardless of the data partitioning. Our approach of secure aggregation using MPC is general enough to support any machine learning algorithm. Our long-term goal is to improve understanding of the utility-privacy trade-off in distributed learning, and provide mechanisms for maximizing utility while satisfying privacy requirements.

**Code:** https://github.com/bargavj/distributedMachineLearning.git

**Acknowledgements.** This work was partially supported by the National Science Foundation (Awards #1111781, #1717950, and #1804603) and research awards from Google, Intel, and Amazon.

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
