[Supplementary Material · supplementary.pdf]

# A   Proofs of the Main Theorems

## A.1   Proof of Theorem 3.2

Before we prove Theorem 3.2, we provide a bound on the Laplace random vector given in below Lemma (as proved in Chaudhuri and Monteleoni [10]).

**Lemma A.1.** *Given a $d$-dimensional random variable $\eta \sim Lap(\beta)$ with $P(\eta) = \frac{1}{2\beta} e^{-\frac{\|\eta\|_1}{\beta}}$, with probability at least $1 - \delta$, the $\ell_2$-norm of the random variable is bounded as $\|\eta\| \leq d\beta \log\left(\frac{d}{\delta}\right)$.*

For any differentiable and convex objective function, Chaudhuri and Monteleoni [10] propose the following lemma to bound the sensitivity of model estimator:

**Lemma A.2** (Lemma 1 of Chaudhuri and Monteleoni [10]). *Let $\theta_1 = \arg\min_\theta G(\theta)$ and $\theta_2 = \arg\min_\theta G(\theta) + g(\theta)$ such that $G(\theta)$ and $g(\theta)$ are both differentiable and convex. Then $\|\theta_1 - \theta_2\| \leq \frac{g_1}{G_2}$ where $G_2 = \min_v \min_\theta v^\top \nabla^2 G(\theta) v$ for any unit vector $v \in \mathbb{R}^d$ and $g_1 = \max_\theta \|\nabla g(\theta)\|$.*

We also give the following theorem to bound the excess risk between distributed non-private model estimator and the optimal model estimator in the centralized setting. This theorem is used to prove the Theorem 3.2.

**Theorem A.3.** *Given an aggregate model estimator, $\widehat{\theta} = \frac{1}{m}\sum_{j=1}^m \widehat{\theta}^{(j)}$ where $\widehat{\theta}^{(j)} = \arg\min_\theta \frac{1}{n_j}\sum_{i=1}^{n_j} \ell(\theta, x_i^{(j)}, y_i^{(j)}) + \lambda N(\theta)$ and an optimal model estimator $\theta^*$ trained on the centralized data such that the data lie in a unit ball and $\ell(\cdot)$ is $G$-Lipschitz , we have*

$$\|\widehat{\theta} - \theta^*\| \leq \frac{G(m-1)}{n_{(1)}\lambda}.$$

*Proof.* For a party $P_j$, the local model estimator is given as:

$$\widehat{\theta}^{(j)} = \arg\min_\theta \frac{1}{n_j}\sum_{i=1}^{n_j} \ell(\theta, x_i^{(j)}, y_i^{(j)}) + \lambda N(\theta) = \arg\min_\theta G(\theta).$$

The centralized model estimator is hence given as:

$$\theta^* = \arg\min_\theta \frac{1}{n_j}\sum_{i=1}^{n_j} \ell(\theta, x_i^{(j)}, y_i^{(j)}) + \sum_{l \neq j}\frac{1}{n_l}\sum_{i=1}^{n_l} \ell(\theta, x_i^{(l)}, y_i^{(l)}) + \lambda N(\theta) = \arg\min_\theta G(\theta) + g(\theta).$$

Thus we have the following values of $g_1$ and $G_2$:

$$g_1 = \max_\theta \|\nabla g(\theta)\| = \max_\theta \sum_{l \neq j}\frac{1}{n_l}\|\nabla\ell(\theta, x_i^{(l)}, y_i^{(l)})\| \leq G\sum_{l \neq j}\frac{1}{n_l}$$

$$G_2 = \min_v \min_\theta \|v^\top \nabla^2 G(\theta) v\| = \min_v \min_\theta \|v^\top(\frac{1}{n_j}\|\nabla^2\ell(\theta, x_i^{(j)}, y_i^{(j)})\| + \lambda.1)v\| \geq \lambda$$

Using Lemma A.2, we have $\|\widehat{\theta}^{(j)} - \theta^*\| = \frac{G}{\lambda}\sum_{l \neq j}\frac{1}{n_l}$. Applying triangle inequality, we get:

$$\|\widehat{\theta} - \theta^*\| \leq \frac{1}{m}\sum_j \|\widehat{\theta}^{(j)} - \theta^*\| = \frac{G}{m\lambda}\sum_j\sum_{l \neq j}\frac{1}{n_l} = \frac{G(m-1)}{m\lambda}\sum_j\frac{1}{n_j} \leq \frac{G(m-1)}{n_{(1)}\lambda}.$$

$\square$

Now we are ready to prove Theorem 3.2 using Lemma A.1 and Theorem A.3.

*Proof of Theorem 3.2.* Using Taylor Expansion, we have

$$J(\widehat{\theta}^{\mathrm{priv}}) = J(\theta^*) + (\widehat{\theta}^{\mathrm{priv}} - \theta^*)\nabla J(\theta^*) + \frac{1}{2}(\widehat{\theta}^{\mathrm{priv}} - \theta^*)\nabla^2 J(\theta)(\widehat{\theta}^{\mathrm{priv}} - \theta^*),$$

where $\theta = \alpha\widehat{\theta}^{\mathrm{priv}} + (1-\alpha)\theta^*$ for some $\alpha \in [0,1]$. By definition, $\nabla J(\theta^*) = 0$, thus we get

$$J(\widehat{\theta}^{\mathrm{priv}}) - J(\theta^*) \le \frac{1}{2}\|\widehat{\theta}^{\mathrm{priv}} - \theta^*\|^2 \cdot \|\nabla^2 J(\theta)\|.$$

Since $\ell(\cdot)$ is $L$-smooth, we have $\|\nabla^2 J(\theta)\| \le \lambda + L$, therefore

$$\begin{aligned}
J(\widehat{\theta}^{\mathrm{priv}}) \le{}& J(\theta^*) + \frac{\lambda+L}{2}\|\widehat{\theta} - \theta^* + \eta\|^2 \\
\le{}& J(\theta^*) + \frac{\lambda+L}{2}\big[\|\widehat{\theta} - \theta^*\|^2 + \|\eta\|^2 + 2(\widehat{\theta} - \theta^*)^\top \eta\big] \\
\le{}& J(\theta^*) + \frac{\lambda+L}{2}\big[\|\widehat{\theta} - \theta^*\|^2 + \|\eta\|^2 + 2\|\widehat{\theta} - \theta^*\| \cdot \|\eta\|\big].
\end{aligned}$$

By Theorem A.3 and Lemma A.1, we obtain

$$\begin{aligned}
J(\widehat{\theta}^{\mathrm{priv}}) \le{}& J(\theta^*) + \frac{G^2(m-1)^2(\lambda+L)}{2n_{(1)}^2\lambda^2} + \frac{2G^2 d^2(\lambda+L)}{m^2 n_{(1)}^2 \lambda^2 \epsilon^2}\log^2\left(\frac{d}{\delta}\right) \\
& + \frac{2G^2 d(m-1)(\lambda+L)}{mn_{(1)}^2\epsilon\lambda^2}\log\left(\frac{d}{\delta}\right) \\
\le{}& J(\theta^*) + C_1\frac{G^2(\lambda+L)}{n_{(1)}^2\lambda^2}\left(m^2 + \frac{d^2\log^2(d/\delta)}{m^2\epsilon^2} + \frac{d\log(d/\delta)}{\epsilon}\right),
\end{aligned}$$

where $C_1$ is an absolute constant. $\qquad\square$

## A.2  Proof of Theorem 3.3

*Proof.* The proof follows from Sridharan et al. [48] where the authors give the following relation between true risk and empirical risk bounds:

$$\widetilde{J}(\widehat{\theta}^{\mathrm{priv}}) - \min_\theta \widetilde{J}(\theta) \le 2\big[J(\widehat{\theta}^{\mathrm{priv}}) - J(\theta^*)\big] + \frac{16G^2}{\lambda n}\left[32 + \log\left(\frac{1}{\delta}\right)\right].$$

Substituting the empirical risk bound from Theorem 3.2 will complete the proof. $\qquad\square$

## A.3  Proof of Corollary 3.5

*Proof.* By composition property of Lemma 2.2, each $\theta_t$ is $(t\rho)$-zCDP.

From Lemma 2.3, the privacy budget $\epsilon_t$ for iteration $t$ is given as $\epsilon_t = t\rho + 2\sqrt{t\rho\log(1/\delta)}$

Total privacy budget is $\epsilon = T\rho + 2\sqrt{T\rho\log(1/\delta)}$

$$\begin{aligned}
\implies \frac{t\epsilon}{T} ={}& t\rho + 2\sqrt{t\rho\log(1/\delta)}\left(\sqrt{\frac{t}{T}}\right) \\
={}& t\rho\left(1 - \sqrt{\frac{t}{T}}\right) + \sqrt{\frac{t}{T}}\epsilon_t \\
\implies \epsilon_t ={}& \sqrt{\frac{t}{T}}\epsilon + \sqrt{Tt}\left(\sqrt{\frac{t}{T}} - 1\right)\rho
\end{aligned}$$

In the proof of Theorem 3.4, we showed that: $\rho \approx \frac{\epsilon^2}{4T\log(1/\delta)}$

Substituting the value of $\rho$, we get the relation between $\epsilon_t$ and $\epsilon$:

$$\epsilon_t = \sqrt{\frac{t}{T}}\epsilon + \sqrt{\frac{t}{T}}\left(\sqrt{\frac{t}{T}} - 1\right)\frac{\epsilon^2}{4\log(1/\delta)} \le \sqrt{\frac{t}{T}}\epsilon$$

Hence, each intermediate model estimator $\theta_t$ is $(\sqrt{t/T}\epsilon, \delta)$-differentially private. $\qquad\square$

## A.4 Proof of Theorem 3.6

*Proof.* From $L$-smoothness assumption:

$$\mathbb{E}[J(\theta_{t+1}) - J(\theta_t)] \leq \mathbb{E}[\langle \nabla J(\theta_t), \theta_{t+1} - \theta_t \rangle + \frac{1}{2L}\|\nabla J(\theta_t) + z_t\|^2]$$

$$= \mathbb{E}[-\frac{1}{L}\langle \nabla J(\theta_t), \nabla J(\theta_t) + z_t \rangle + \frac{1}{2L}\|\nabla J(\theta_t) + z_t\|^2]$$

$$= -\frac{1}{2L}\|\nabla J(\theta_t)\|^2 + \frac{1}{2L}\mathbb{E}_{z_t}\|z_t\|^2 \leq -\frac{\lambda}{L}(J(\theta_t) - J^*) + \frac{d\sigma^2}{2L}$$

The last inequality comes from the strong convexity assumption. The above equation is conditioned on the randomness of $\theta_t$, and can be written as:

$$\mathbb{E}[J(\theta_{t+1})] - J(\theta^*) \leq (1 - \frac{\lambda}{L})(J(\theta_t) - J(\theta^*)) + \frac{d\sigma^2}{2L}$$

Summing over $t = 0, \ldots, T$ iterations, and taking expectation:

$$\mathbb{E}[J(\theta_T)] - J(\theta^*) \leq (1 - \frac{\lambda}{L})^T (J(\theta_0) - J(\theta^*)) + \frac{d\sigma^2}{2\lambda}$$

When $T = O\Big( \log \big( \frac{m^2 n_{(1)}^2 \epsilon^2}{dG^2 \log(1/\delta)} \big) \Big)$,

$$\mathbb{E}[J(\theta_T)] - J(\theta^*) \leq C_1 \frac{G^2 d \log^2(mn_{(1)}) \log(1/\delta)}{m^2 n_{(1)}^2 \epsilon^2}$$

where $C_1$ is an absolute constant and the big-$O$ notation hides other $\log, L, \lambda$ terms. □

## A.5 Proof of Theorem 3.7

*Proof.* We want to bound $\mathbb{E}[\widetilde{J}(\theta_T)] - \min_\theta \widetilde{J}(\theta)$, where $\widetilde{J}(\theta) = \mathbb{E}_D[J_D(\theta)]$. We denote $\widetilde{J}(\widetilde{\theta}) = \min_\theta \widetilde{J}(\theta)$. According to Theorem 1 in [48], we have the following holds with probability at least $1 - \gamma$

$$\widetilde{J}(\theta^T) - \widetilde{J}(\widetilde{\theta}) \leq 2(J(\theta^T) - J(\theta^*)) + C_2 \frac{G^2 \log(1/\gamma)}{\lambda m n_{(1)}},$$

where $C_2$ is an absolute constant. Therefore, we have the following holds with probability at least $1 - \gamma$

$$\mathbb{E}[\widetilde{J}(\theta_T)] - \min_\theta \widetilde{J}(\theta) \leq 2(\mathbb{E}[J(\theta_T)] - J(\theta^*)) + C_2 \frac{G^2 \log(1/\gamma)}{\lambda m n_{(1)}}$$

$$\leq C_1 \frac{G^2 d \log^2(mn_{(1)}) \log(1/\delta)}{m^2 n_{(1)}^2 \epsilon^2} + C_2 \frac{G^2 \log(1/\gamma)}{\lambda m n_{(1)}},$$

where $C_1$ is an absolute constant. □

# B More Experimental Results

## B.1 Experiments on KddCup99 dataset

With the increasing number of parties $m$ (and accordingly decreasing the local data set size $n_{(1)}$), performance of all the methods decrease except that of MPC Grad P (see Figures 3 and 4). While the performance of baselines deteriorate mainly due to the large amount of noise they add, the performance of MPC Out P, on the other hand, decreases with decreasing local data set size due to the loss in information from data partitioning (which is the case with any model aggregation based method, including the non-private ones). For $m = 50,000$, the large amount of noise destroys the utility of Pathak (Figure 4).

Figure 3: Optimality Gap and Relative Accuracy Loss Comparison on KddCup99 ($m = 100$)
*All models except Rajkumar and Agarwal have privacy budget $\epsilon = 0.5$.

Figure 4: Optimality Gap and Relative Accuracy Loss Comparison on KddCup99 ($m = 50,000$)
*All models except Rajkumar and Agarwal have privacy budget $\epsilon = 0.5$.

Figure 5: Optimality Gap and Relative Accuracy Loss Comparison on KddCup98 ($m = 100$)
*All models except Rajkumar and Agarwal have privacy budget $\epsilon = 0.5$.

Figure 6: Optimality Gap and Relative Accuracy Loss Comparison on KddCup98 ($m = 50,000$)
*All models except Rajkumar and Agarwal have privacy budget $\epsilon = 0.5$.

Figure 7: Optimality Gap and Relative Accuracy Loss Comparison on Adult ($m = 100$)
*All models except Rajkumar and Agarwal have privacy budget $\epsilon = 0.5$.

### B.2 Experiments on KddCup98 dataset

Overall, with the increasing number of parties $m$ (and accordingly decreasing the local data set size $n_{(1)}$), performance of all the methods decrease except that of MPC Grad P (see Figures 5 and 6). Though the performance of MPC Out P decreases with decreasing the local data set size, it still outperforms the baselines of model aggregation. We note that for $m = 50,000$, Local Obj P performs worse than the Local Out P which is due to the deviation in the objective function of Local Obj P as mentioned earlier. The utility of Pathak is severely affected due to the large amount of noise added, which is why the plot is out of the range for $m = 50,000$ (Figure 6).

### B.3 Experiments on Adult dataset

The Adult [2] data set consists of demographic information of approximately 47,000 individuals, and the task is to predict whether the annual income of an individual is above or below $50,000. After removing records with missing values, we end up with 45,222 records of which 30,000 records formed the training set and the remaining records formed the test set. After pre-processing, each record consisted of 104 features.

We vary the number of parties $m$ from 100 (where each party has 300 data instances) to 1,000 parties (with each party having 30 data instances) and all the way to 30,000 parties (each having only one data instance).

Our proposed methods MPC Out P and MPC Grad P outperform the baselines both in terms of optimality gap and relative accuracy loss (Figures 7, 8 and 9). MPC Grad P achieves optimality

Figure 8: Optimality Gap and Relative Accuracy Loss Comparison on Adult ($m = 1,000$)
*All models except Rajkumar and Agarwal have privacy budget $\epsilon = 0.5$.

Figure 9: Optimality Gap and Relative Accuracy Loss Comparison on Adult ($m = 30,000$)
*All models except Rajkumar and Agarwal have privacy budget $\epsilon = 0.5$.

gap in the order of $10^{-2}$ and relative accuracy loss in the order of $10^{-4}$ for all settings, while the performance of MPC Out P deteriorates with decreasing local data set size (due to the information loss from data partitioning). Nevertheless, MPC Out P still outperforms all the baselines by orders of magnitude.

## C    Implementation of Secure Aggregation

For our prototype implementation of secure aggregation, we implemented the two-party non-collusive framework of Tian et al. [49]. We used the Obliv-C [57] framework for performing the MPC, and measured the runtime and bandwidth cost of doing secure aggregation along with noise generation inside the MPC framework (See Table 2). We conducted experiments using both semi-honest Yao's garbled circuits protocols, and in the active-secure dual execution model [28]. This provides security against fully malicious adversaries, but leaks up to one arbitrary bit of information about private inputs with each protocol execution.

Across all the data sets, the garbled circuit for performing the secure aggregation along with noise generation took around 700 to 900 MB of bandwidth, consisting of around 37 million to 48 million gates for Yao's protocol. The primary cost of the protocol is transmitting the ciphertexts of the garbled gates, and the circuit complexity is dominated by the noise generation. The bandwidth of oblivious transfer (OT) ranged from 32 to 41 MB. The cost is double for dual execution protocol. Note that the cost does not depend on the size of the data set; it only depends on the number of parties, the number of features in the data set, the needed precision, and the method used to add the required differential

Table 2: Secure Aggregation in MPC

|  | Adult | KDDCup99 | KDDCup98 |
|---|---|---|---|
| Number of Features | 104 | 122 | 95 |
| Gate Count | 40,733,500 | 47,781,200 | 37,209,600 |
| GC Bandwidth (MB) | 776.93 | 911.35 | 709.72 |
| OT Bandwidth (MB) | 35.17 | 41.19 | 32.28 |
| Yao Runtime (sec) | 22.28 | 26.45 | 20.31 |
| DualEx Runtime (sec) | 56.46 | 65.21 | 50.67 |

privacy noise. As the table suggests, the number of gates grows linearly with the number of features, with roughly 391,600 gates per feature.

Overall secure aggregation took less than 2 seconds using Yao's protocol, and the remaining computation time was taken by the noise generation. For sampling the Laplace noise, each party first inputs its share of random number $u_i$, which is XORed inside the MPC to generate the randomness $u = \bigoplus u_i$. Next, the Laplace noise of scale $b$ is generated as: $\text{Lap}(b) = \pm \log([u]).b$, where $[.]$ shrinks the input to (0,1] range and $b = \frac{2}{m.n_{(1)}\lambda\epsilon}$. We used the big integer library [15] to perform the log operation within the MPC. Gaussian noise can be generated in the same way using the Box-Muller method [5]. There are many opportunities to reduce this cost without reducing security by generating the required noise differently, which we plan to explore in future work.