[Reviews · NeurIPS 2018]

Reviewer 1



This paper proposes a multiparty learning algorithm with two different approaches. In the first approach, they combine local models from different owners and then add noise to the global model before revealing it. In the second approach, the global model is trained by each data owner jointly and in each iteration, a sufficient amount of noise is added to the gradient. The gradient perturbation method proposed in this paper is similar to the method proposed in “A Differentially Private Stochastic Gradient Descent Algorithm for Multiparty Classification” by Rajkumar and Agarwal. First of all, it would be nice to mention the differences between that work. I think it’s also necessary to use that approach as a baseline in the experiments section. One general problem of this paper is referring to the appendix a lot. This makes the paper harder to follow and understand. Maybe, some of the proofs can be included in the main part of the paper and the more complicated mathematical operations can be given in the appendix. This situation affects the comprehensiveness of the paper. I also have some doubts about the “G” parameter. Is it possible to set the Lipschitz constant to 1 in this case? If it is how do you prove that? Other than this the number of iterations is important in differentially private settings and affect the performance a lot. How did you choose the “T”? In most of the cases MPC Grad P outperforms the other algorithms. How do you explain it performs better than the MPC Output P? To sum up, this is an important problem and this paper brings a solution to this problem.

Reviewer 2



The paper is about learning in a decentralized setting where parties have their own dataset and learn privately a global model. Privacy is usually guaranteed either by cryptographic procedures (secure multi party computation, MPC) or by introducing noise and theoretically analyzed in the framework of differential privacy (DP). Following Pathak et al [39], the paper combines MPC and DP. But the main contribution is to reduce the noise by a factor sqrt(m) using two different techniques: one from Chauduri et al [10] and one from Bun and Steike [6]. The paper considers two settings, output perturbation and gradient perturbation. In the output perturbation scheme, the parties securely send their model learned locally to a central server, the server aggregates (averages) the models to build a joined model and then adds noise before releasing it. The average is privately computed in the MPC protocol. The approach is therefore very similar to the one of Pathak. The contribution in this part of the paper is to use a bound proved in Chauduri et al [10] in the proof that evaluates the needed noise to be \epsilon-DP. In the gradient perturbation scheme, noise is introduced at every gradient step, after averaging the gradients, and before sending a new model back to the parties. Therefore, a general result is obtained using composition results in DP. The paper relies on zero concentrated DP for a more efficient combination step (Bun and Steike [6]). The new bounds improves also the impact of noise in the true and empirical risks (and therefore it is important for associated ERM problems). Most of the proofs are not original (adapted from Chauduri and Monteleoni [9]) but easy to follow and clearly exposed. The paper is clearly written. The authors have correctly explained the motivations and have honestly indicated at the core of their contribution, to whom this or that result was attributed. Detailed proofs are given in the appendix. The authors claim in different points of the paper that the benefit is explained by the fact that they add noise inside the MPC. This is not clear for me because in each scheme, the noise is introduced after model aggregation (so it could be "after" rather than "inside"). In conclusion, the paper improves the known bounds and shows theoretically and experimentally the benefits of the proposed algorithms. The significance is moderated by the fact that the results are obtained from small adaptations of the known techniques using each time another work. But I think that the paper is clearly written and the results are interesting. Technical: - l133-134 is difficult to understand (almost false). I think that the authors want to say that Z is concentrated around 0 and then it becomes unlikely to distinguish D from D'. - I am not sure that the comparison with Pathak et al is fair enough because they use the l1 norm to bound the difference between two models learned from closeby by datasets UPDATE: "The key difference is whether the noise is added before or after the models are aggregated." Indeed, you're right and it is more clear to say after r before aggregation than "inside/within" (because the MPC is not really modified).

Reviewer 3



The problem under investigation in this work is jointly training of convex models over private data. The authors propose to combine MPC with differential privacy. By using MPC the amount of noise required by the differential privacy procedure can be reduced and therefore more accurate models can be built. ======= After reading the response of the authors my evaluation of this work has increased. I do see that there is some novelty in this work but given that several key references were missing, the paper requires a major revision. Moreover, the amount of novelty is limited compared to what is stated in the current version. While this paper may be accepted, it may have larger impact if the authors would take the time to make a big facelift to the paper as well as add more novel content. I have several concerns about this work: 1. The novelty of this work is not clear. For example, Chase et al.: “Private Collaborative Neural Network Learning” have already proposed using MPC to reduce the amount of noise required by Differential Privacy. They proposed a method that scales the noise by the size of the joint dataset, instead of the size of the smallest dataset $n_1$ by jointly computing the some of gradients and the number of records instead of each party computing the average on its data. 2. From the statements of the theorems it is not clear if the approach described here provides the right level of privacy. For example, Theorem 3.4 states that \Theta_T is \epsilon,\delta-differentially private. However, the participants in the joint learning process see \Theta_1, \Theta_2,\ldots,\Theta_T. What do these participants learn about other participants? 3. When adding the noise in the MPC setting it is important to define how the noise is generated since the party that generates the noise should not learn about the data more than allowed by the privacy requirements. 4. In theorem 3.5 the learning rate is bonded to the smoothness of the loss function, why does it make sense?